# Impact of exercise training associated with enalapril treatment on blood pressure variability and renal dysfunctions in an experimental model of arterial hypertension and postmenopause

**Michel Pablo dos Santos Ferreira Silva[1], Maycon Junior Ferreira[2], Tânia Plens Shecaira[1,2], Danielle da Silva Dias[2,3,4], Débora Conte Kimura[5], Maria Cláudia Irigoyen[4], Guiomar Nascimento Gomes[5], Kátia De Angelis**[1,2]*

1 Translational Physiology Laboratory, Universidade Nove de Julho (UNINOVE), São Paulo, SP, Brazil,
2 Exercise Physiology Laboratory, Universidade Federal de São Paulo (UNIFESP), São Paulo, SP, Brazil,
3 Hypertension Unit, Heart Institute (InCor), School of Medicine, Universidade de São Paulo (USP), São Paulo, SP, Brazil, 4 Postgraduate Program in Physical Education, Universidade Federal do Maranhão, São Luís, MA, Brazil, 5 Department of Physiology, Universidade Federal de São Paulo (UNIFESP), São Paulo, SP, Brazil

* prof.kangelis@yahoo.com.br

## Abstract

### Objective

In this study, we aimed to investigate the effects of the concurrent exercise training (CET) associated with the enalapril maleate on blood pressure variability (BPV) and renal profile in an experimental model of arterial hypertension (AH) and postmenopause.

### Methods

Female ovariectomized spontaneously hypertensive rats (SHR) were distributed into 4 groups (n = 8/group): sedentary (SO), sedentary + enalapril (SOE), trained (TO) and trained + enalapril (TOE). Both enalapril (3mg/kg) and CET (3 days/week) were conducted during 8 weeks. Blood pressure (BP) was directly recorded for BPV analyses. Renal function, morphology, inflammation and oxidative stress were assessed.

### Results

The SOE, TO e TOE groups presented decreased systolic BP compared with SO. Both trained groups (TO and TOE) presented lower BPV and increased baroreflex sensitivity (TO: $0.76 \pm 0.20$ and TOE: $1.02 \pm 0.40$ vs. SO: $0.40 \pm 0.07$ ms/mmHg) compared with SO group, with additional improvements in TOE group. Creatinine and IL-6 levels were reduced in SOE, TO and TOE compared with SO group, while IL-10 was increased only in TOE group (vs. SO). Enalapril combined with CET promote reduction in lipoperoxidation (TOE: $1.37 \pm 0.26$ vs. SO: $2.08 \pm 0.48$ and SOE: $1.84 \pm 0.35$ μmol/mg protein) and hydrogen peroxide (TOE: $1.89 \pm 0.40$ vs. SO: $3.70 \pm 0.19$ and SOE: $2.73 \pm 0.70$ μM), as well as increase in

**Data Availability Statement:** All relevant data are within the manuscript.

**Funding:** FAPESP (2019/06277-0), São Paulo Research Foundation (https://fapesp.br/). National Council for Scientific and Technological Development (CNPq) (406792/2022-4) (https://www.gov.br/cnpq/pt-br). Maycon Junior Ferreira is a beneficiary of the FAPESP Scholarship, while Kátia De Angelis and Maria Claudia Irigoyen are recipients of the CNPq Fellowship (CNPq-BPQ). Please note that the funders played no role in the study's design, data collection and analysis, decision to publish, or manuscript preparation.

**Competing interests:** The authors have declared that no competing interests exist.

catalase activity (vs. sedentary groups). The tubulointerstitial injury was lower in interventions groups (SOE, TO and TOE vs. SO), with potentialized benefits in the trained groups.

## Conclusions

Enalapril combined with CET attenuated BPV and baroreflex dysfunctions, probably impacting on end-organ damage, as demonstrated by attenuation in the AH-induced renal inflammations, oxidative stress and morphofunctional impairments in postmenopausal rats.

## Introduction

Arterial hypertension (AH) is the main cause of chronic kidney disease (CKD) and the association of this two conditions exacerbates considerably the cardiovascular risk [1], once the mortality by CVD increases proportionally due to chronic kidney disease severity [2]. The chronic BP elevation predisposes to atherosclerosis, which in turn may result in renovascular AH and/or ischemic nephropathy [3, 4].

BP variability (BPV) has been well documented as an important and independent risk factor for acute kidney injury [5] and its oscillation is related to the development of cardiac, vascular and kidney tissue damages [6]. In turn, recent evidence showed that low glomerular filtration rates was related to high BPV in medicated hypertensive patients [7]. The mechanisms involved in the increase in BPV are not completely elucidated yet. Importantly, we have showed a relationship between increased BPV and lipoperoxidation [8], suggesting that the reduction of BP oscillation may contribute to the prevention of target organ damage.

The inflammatory process in AH is associated with an increased risk of developing cardiovascular events [9]. Low-grade inflammation in the kidneys is the result of an interaction between angiotensin II activity, oxidative stress and sympathetic activity. Moreover, the sympathetic nervous system directly favors the increase of pro-inflammatory cytokines, contributing to the increase in BP [10].

On the other hand, the antihypertensive treatment with angiotensin converting enzyme (ACE) inhibitors has been recommended as an initial therapy in the treatment of AH and enalapril maleate is recognized as one of the antihypertensive drugs most effective for BP reduction [11]. Experimental study with hypertensive rats treated with enalapril showed reduction in cardiac oxidative stress, suggesting protective effects on organs impacted by increased BP [12]. Furthermore, recent meta-analysis demonstrated that exercise is equally efficient in reducing systolic BP (SBP) when compared to antihypertensive drugs [13]. Our group has shown improvement of ventricular function, in addition reduction of protein damages and inflammatory markers in cardiac tissue in diabetic ovariectomized rats submitted to combined exercise training [14]. In addition, we showed that aerobic and resistance exercise training promoted reduction in mean BP (MBP), heart rate (HR), cardiac sympathetic modulation and inflammation, as well as reduced lipoperoxidation and increased antioxidant capacity in renal tissue in ovariectomized hypertensive rats [15].

Considering that the increase in BP in females after menopause encompasses a complex etiology with numerous contributing factors [16] and that both pharmacologic and nonpharmacologic approaches are recommended to AH treatment [17], we hypothesize that the enalapril plus concurrent exercise training (CET) treatment could optimize hemodynamic and autonomic profiles, attenuating renal injury in an experimental model of AH and postmenopause. Therefore, this study sought to investigate the effects of enalapril maleate treatment combined

with CET on BPV, inflammation and morphological parameters in kidney tissue in hypertensive rats submitted to ovarian hormone deprivation. We hypothesize that the combination of these approaches can result in a better autonomic and inflammatory adjustment, resulting in prevention of end-organ damage.

## Material and methods

Female *spontaneously hypertensive rats* (SHR) (150–200 grams, 90 days old) were obtained from Nove de Julho University (UNINOVE). The rats were distributed randomly into 4 groups (n = 7−8 animals/group): sedentary ovariectomized (SO), sedentary ovariectomized treated with enalapril (SOE), trained ovariectomized (TO) and trained ovariectomized treated with enalapril (TOE). The rats were maintained in a temperature-controlled room (22–25˚) under a 12/12−h dark/light cycle. This protocol was approved by Ethics Committee from UNIFESP (CEUA) (n˚ 7611290618) and conducted in agreement with the Guide for the Care and Use of Laboratory Animals [18].

### Ovariectomy

The rats were ovariectomized at 90 days of age as described previously [19]. The rats were anesthetized (80 mg/kg ketamine and 12 mg/kg xylazine, *ip*), and a small abdominal incision was performed. The oviduct was sectioned and the ovaries were removed. The muscle wall and skin were then sutured with nylon and silk thread, respectively. An *im* dose of penicillin (10000 U/kg) was administered after surgery. Ovariectomy was associated with estradiol levels below detectable limits by immunoassay methods [20].

### Interventions

**Pharmacological treatment.** Enalapril maleate (3 mg·kg$^{-1}$·day$^{-1}$) was administered orally (dissolved in drinking water) for 5 days (adaptation phase). Following this, the SOE and TOE groups were treated with the medication for 8 weeks. Enalapril was chosen as the antihypertensive medication based on its proven efficacy in reducing BP [21] and its clinical use [22].

**Exercise training.** Maximal running and maximal load tests were performed on the treadmill and ladder adapted for rats to evaluate exercise capacity of all groups and prescription of exercise training (groups TO e TOE) according to previously described in details [23, 24].

CET was composed by training sessions of aerobic and resistance exercises performed at the same day, with frequency of 3 sessions/week during 8 weeks. The sessions were initiated by aerobic exercise (30–40 min) followed by resistance exercise (15–20 climbs per session, a 1-minute time interval between climbs [14, 15]. Both exercises were performed in moderate intensity (40–60% of the maximum capacity achieved previously in exercises tests) [19].

### Renal function assessment

For evaluation of renal function, the rats were kept in metabolic cages (Nalgene, Ugo Basile, Italy) at the end of the seventh week per 24 hours, where they were weighed at the beginning and at the end, with free access to water and food during this period. Urinary and faecal debt, as well as water and food intake were measured. The urine sample was stored in freezer at -80˚C for later measurement of creatinine and urea levels.

Creatinine levels in plasma and urine were measured by Jaffé method. The accumulation of urea in plasma was determined with an enzymatic colorimetric method using a specific kit (Labtest Diagnostics, Lagoa Santa, Brazil). The measurement of proteins in the urine followed the method recommended by Bradford [25].

## Hemodynamic and autonomic assessments

Hemodynamic and autonomic assessments were conducted at the end of the protocol.

The rats were anesthetized (*ip*) with ketamine (80 mg/kg) and xylazine (12 mg/kg) and cannulas were implanted into the carotid artery toward the left ventricle for direct BP recording [23]. On the next day after catheterization, rats were connected to an extension of 20 cm (PE-50). This extension was connected to an electromagnetic transducer (Blood Pressure XDCR, Kent Scientific, Litchfield, CT, EUA) connected to a preamplifier (Stemtech BPMT-2, Quintron Instrument Inc, Milwaukee, EUA). BP was recorded during 30 minutes in a microcomputer equipped with an analogical digital converter (Windaq, 2hKz, DATAQ Instruments, Akron, OH, EUA), allowing analysis of pulse intervals, beat-to-beat [19].

Standard deviation (SD) from the mean of three-time series of 5 min for each rat was used to obtain the pulse interval (PI) and SBP variability in time-domain. For frequency domain analysis, the same time series of PI and SBP were cubic spline interpolated (250 Hz) and cubic spline decimated to be equally spaced in time after linear trend removal; power spectral density was obtained through the Fast Fourier Transformation.

Spectral power for low-frequency (LF, 0.20–0.75 Hz) band was calculated by power spectrum density integration within each frequency bandwidth, using a customized routine (MATLAB 6.0, Mathworks). The coherence between the PI and SBP signal variability was assessed through cross-spectral analysis [26]. The alpha index was calculated only when the magnitude of the squared coherence between the PI and SBP signals exceeded 0.5 (range 0–1) in the LF band. After coherence calculation, the alpha index was obtained from the square root of the ratio between PI and SBP variability in the LF two major bands [15, 27].

## Tissue collection

The rats were pre-anesthetized with ketamine one day after hemodynamic evaluations and were submitted to euthanasia by decapitation. Both kidneys were immediately removed after euthanasia and properly weighed. Right and left kidneys were maintained in Bouin (for histological analysis) and immediately frozen at -80˚C (for inflammatory profile and oxidative stress analyses), respectively.

## Renal histology

After washed in saline solution, the right kidney was maintained in Bouin solution for 24 hours. After the fixation period, the pieces were dehydrated, diaphanized and paraffinized following the methodology recommended by Michalany [28]. Histological sections about 5 μm thick were stained using Masson trichrome technique for general morphology analyses, such as tissue damage and glomerular quantification.

The glomerular counting area was determined by computerized morphometry (Nikon, NIS-Elements), with 20 fields being analysed on each slide (20x objective). Each field had area of 277.000 μm$^2$. The percentage of the area with morphological changes in the renal interstitium was estimated using image analysis software Image J (NIH) [29]. In the tubule-interstitial morphological analysis, the fields were classified into three morphological alteration ranges: 0 to 25%, 26 to 50% and 51 to 100% of alteration in the evaluated field.

## Inflammatory mediators analyses

Interleukin 6 (IL-6), interleukin 10 (IL-10), and tumor necrosis factor alpha (TNF-α) levels were determined in left renal tissue using a commercially available ELISA kit (R&D Systems Inc.), in accordance with the manufacturer's instructions. ELISA method was performed in

96-well polystyrene microplate with a specific monoclonal antibody coating. Absorbance was measured at 540 nm in a microplate reader.

## Oxidative stress evaluations

**Tissue preparation.** Left renal tissue were cut into small pieces, placed in ice-cold buffer, and homogenized in an ultra-Turrax blender with 1 g of tissue per 5 mL of 120 mmol/L KCl and 30 nmol/L phosphate buffer, pH 7.4. Homogenates were centrifuged for 10 min at 4˚C. Protein was determined as described previously [30].

**Lipoperoxidation–Thiobarbituric Acid Reactive Substances (TBARS).** For the TBARS assay, trichloroacetic acid (10%, w/v) was added to the homogenate to precipitate proteins and to acidify the samples. This mixture was then centrifuged (3.000 rpm, 3 min), the protein-free sample was extracted, and thiobarbituric acid (0.67%, w/v) was added to the reaction medium. The tubes were placed in a water bath (100˚C) for 15 min. Absorbance was measured at 535 nm using a spectrophotometer. A commercially available malondialdehyde was used as a standard, and the results are expressed as µmol/mg of protein [31].

**Protein oxidation–Carbonyls.** The protein damage was determined by protein carbonyls measurements, using 200 mL of sample. Samples were incubated with 2,4-dinitrophenylhydrazine (DNPH 10 mM) in a 2.5M HCl solution for 1h at room temperature in the dark. Samples were vortexed every 15 min. Subsequently, a 20% trichloroacetic acid (w/v) solution was added and the solution was incubated on ice for 10 min and centrifuged for 5 min at 2000 rpm to collect protein precipitates. An additional wash was performed with 10% trichloroacetic acid (w/v). The pellet was washed three times with ethanol/ethyl acetate (1:1) (v/v). The final precipitates were dissolved in 6M guanidine hydrochloride solution and incubated for 10 min at 37˚C, and the absorbance was measured at 360 nm [32].

**Hydrogen peroxide ($H_2O_2$).** The assay was based on the horseradish peroxidase (HRPO)-mediated oxidation of phenol red by $H_2O_2$, leading to the formation of a compound measurable at 610 nm. Kidney homogenates were incubated for 30 min at 37˚C in 10 mmol/l phosphate buffer consisting of 140 mmol/l NaCl and 5 mmol/l dextrose. The supernatants were transferred to tubes with 0.28 mmol/l phenol red and 8.5 U/ml HRPO. After 5 min incubation, 1 mol/l NaOH was added and it was read at 610 nm. The results were expressed in nanomole $H_2O_2$ per gram tissue [33].

**Antioxidant enzymes.** Superoxide dismutase (SOD) activity was measured spectrophotometrically by the rate inhibition of pyrogallol auto-oxidation at 420 nm [34]. Enzyme activity was reported as USOD/mg protein. Catalase (CAT) concentration was measured by monitoring the decrease in $H_2O_2$ concentration at 240 nm, and the results are reported as nmol of $H_2O_2$/mg protein [35].

## Statistical analysis

Data are reported as mean ± standard deviation. Distribution of data was assessed and to test the assumption of homogeneity of variance was used Levene's test. One-way analysis of variance (ANOVA) followed by Student-Newman-Keuls post-hoc test was used to compare groups. Significance level was established at $p \leq 0.05$.

## Results

### General evaluations

There were no differences between groups for body weight, right and left kidney weight at the end of the protocol (Table 1). After 8 weeks of protocol, the trained groups presented a higher

**Table 1. Body weight, renal weight and maximal exercise tests in the studied groups.**

|  | SO | SOE | TO | TOE |
|---|---|---|---|---|
| Final body weight (g) | 231 ± 16 | 238 ± 24 | 231 ± 9 | 233 ± 8 |
| Right kidney (g) | 0.68 ± 0.09 | 0.69 ± 0.06 | 0.65 ± 0.02 | 0.65 ± 0.03 |
| Left kidney (g) | 0.67 ± 0.09 | 0.66 ± 0.06 | 0.67 ± 0.03 | 0.66 ± 0.03 |
| Treadmill final test (minutes) | 23.06 ± 1.22 | 22.30 ± 1.44 | 27.43 ± 3.12*# | 29.56 ± 1.44*#§ |
| Ladder final test (% body weight) | 81.4 ± 17.0 | 87.9 ± 18.2 | 108.8 ± 15.5*# | 142.5 ± 17.9*#§ |

Data are presented as mean ± standard deviation.

SO, sedentary ovariectomized; SOE, sedentary ovariectomized treated with enalapril; TO, trained ovariectomized; TOE, trained ovariectomized treated with enalapril.

\* $p \leq 0.05$ vs. SO

\# $p \leq$ vs. SOE

§ $p \leq 0.05$ vs. TO.

performance in both tests compared to the sedentary groups. Besides that, trained group treated with enalapril presented a higher performance compared to only trained group in both final tests.

## Hemodynamic and BPV evaluations

The groups SOE, TO and TOE presented lower values of SBP compared to the group SO after 8 weeks of protocol (Fig 1A).

In the BPV in the time domain, the trained groups showed significant reductions when compared to the SO group in the SD of SBP (SD-SBP) (TO: 5.9 ± 1.4 and TOE: 5.9 ± 0.6 vs. SO: 7.2 ± 1.0 mmHg) and variance of SBP (VAR-SBP) (TO: 36.1 ± 15.5 and TOE: 32.9 ± 6.4 vs. SO: 52.4 ± 11.5 $mmHg^2$), while any differences were observed in in the SD-SBP between sedentary group treated with enalapril (SOE: 6.8 ± 0.9 $mmHg^2$) compared to the other groups. Additionally, the TOE group also presented lower VAR-SBP in relation to their respective sedentary group. (TOE: 32.9 ± 6.4 vs. SOE: 47.8 ± 10.5 $mmHg^2$) (Fig 1A–1C).

In the frequency domain, the trained groups presented lower LF component of SBP (LF-SBP) compared to the SO group (TO: 7.4 ± 2.4 and TOE: 7.4 ± 3.2 vs. SO 13.4 ± 3.4 SOE: 11.4 ± 4.5 $mmHg^2$) (Fig 1D). The calculation of spontaneous baroreflex sensitivity (alpha index) showed that the trained groups showed an increase compared to the SO group (TO: 0.76 ± 0.20 and TOE: 1.02 ± 0.40 vs. SO: 0.40 ± 0.07 ms/mmHg). Furthermore, trained group treated with enalapril presented higher alpha index in relation to the SOE and TO groups (TOE: 1.02 ± 0.40 vs. SOE: 0.57 ± 0.20 and OT: 0.76 ± 0.20 ms/mmHg) (Fig 1E).

## Renal function assessment

There were no differences in water intake, urinary debt, food intake, and faeces production between the groups when submitted to the metabolic cages (Table 2).

Similarly, no differences were observed for plasmatic urea concentrations (SO: 33.8 ± 5.9, SOE: 35.0 ± 7.1, TO: 32.3 ± 10.2 and TOE: 28.1 ± 6.0 mg/dL). However, plasmatic creatinine levels in SOE (0.32 ± 0.10 mg/dL), TO (0.36 ± 0.05 mg/dL) and TOE groups (0.30 ± 0.03 mg/dL) were reduced when compared to the SO group (SO: 0.49 ± 0.06 mg/dL). There was no difference between groups for creatinine clearance (SO: 6.0 ± 1.2 ml/min; SOE 7.2 ± 2.7 ml/min; TO 8.5 ± 2.5 ml/min and TOE: 9.0 ± 3.7), as well as proteinuria levels (SO: 22.9 ± 4.3; SOE: 19.1 ± 5.1; TO: 18.0 ± 3.3 and TOE: 18.2 ± 3.2 mg/24h) (Fig 2A–2D).

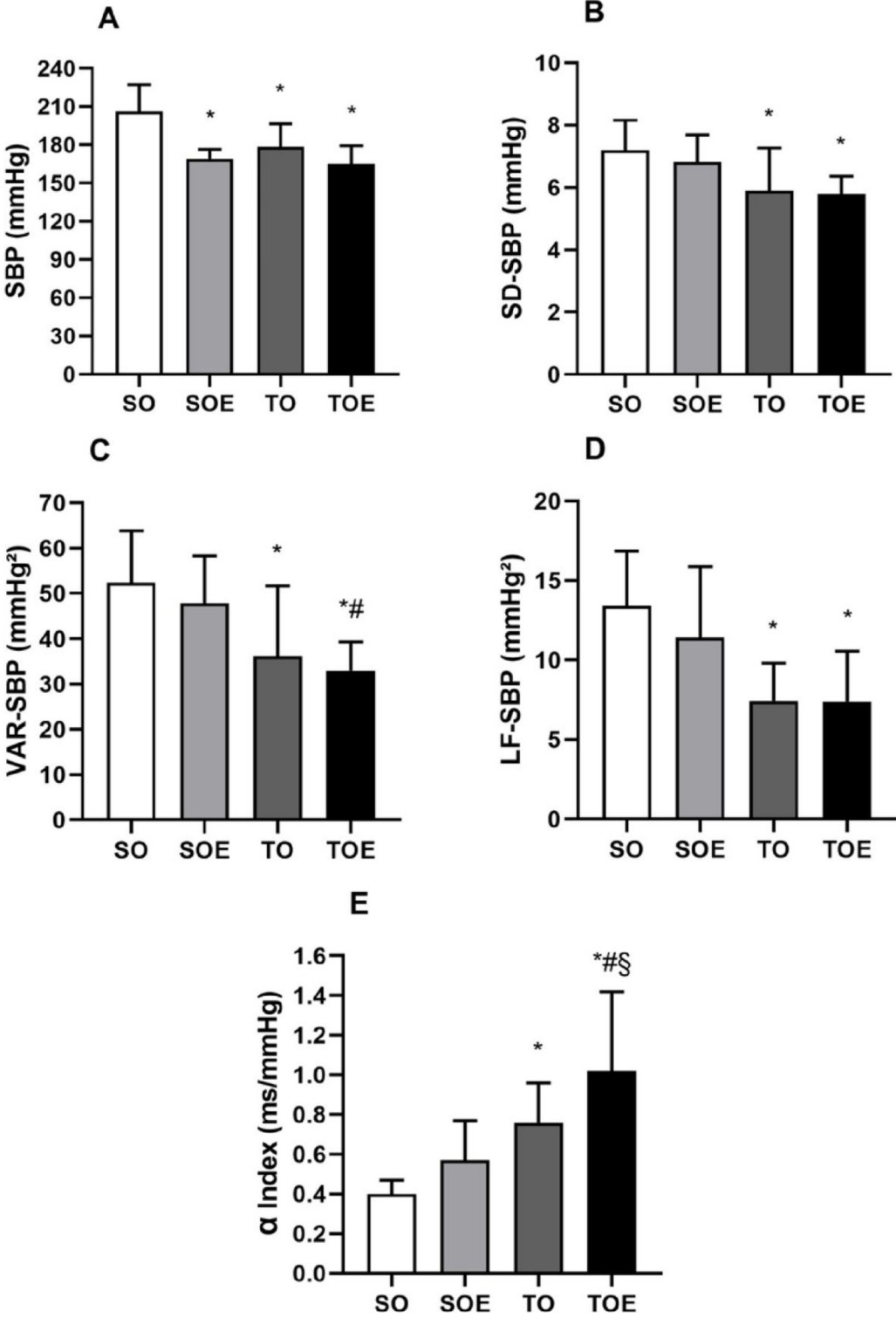

**Fig 1. Blood pressure and blood pressure variability.** (A) Systolic blood pressure, (B) standard deviation of systolic blood pressure, (C) variance of systolic blood pressure, (D) low frequency band of systolic blood pressure and (E) alpha index in sedentary ovariectomized (SO), sedentary ovariectomized treated with enalapril (SOE), trained ovariectomized (TO) and trained ovariectomized treated with enalapril (TOE) groups. Data are presented as mean ± standard deviation. * p ≤ 0.05 vs. SO; # p ≤ 0.05 vs. SOE; $ p ≤ 0.05 vs. TO. SBP, systolic blood pressure; SD-SBP, standard deviation of systolic blood pressure; VAR-SBP, variance of systolic blood pressure; LF-SBP, low frequency band of systolic blood pressure; α, alpha index.

**Table 2. Water and feed intake, urine and faeces volume in 24 hours of metabolic cage in the studied groups.**

|  | SO | SOE | TO | TOE |
|---|---|---|---|---|
| Water (ml) | 33 ± 10 | 34 ± 13 | 32 ± 6 | 34 ± 4 |
| Urine (ml) | 23 ± 6 | 25 ± 9 | 20 ± 6 | 26 ± 5 |
| Feed (g) | 11 ± 3 | 14 ± 3 | 13 ± 4 | 14 ± 3 |
| Faeces (g) | 6 ± 2 | 6 ± 2 | 7 ± 3 | 6 ± 1 |

Data are presented as mean ± standard deviation.

SO, sedentary ovariectomized; SOE, sedentary ovariectomized treated with enalapril; TO, trained ovariectomized; TOE, trained ovariectomized treated with enalapril.

## Renal morphometry and morphology analysis

No differences were found between the groups for glomerular quantification (SO: 71.5 ± 6.4, SOE: 52.0 ± 9.9, TO: 56.0 ± 11.3 and TOE: 65.7 ± 14.0 n° glomeruli/slide).

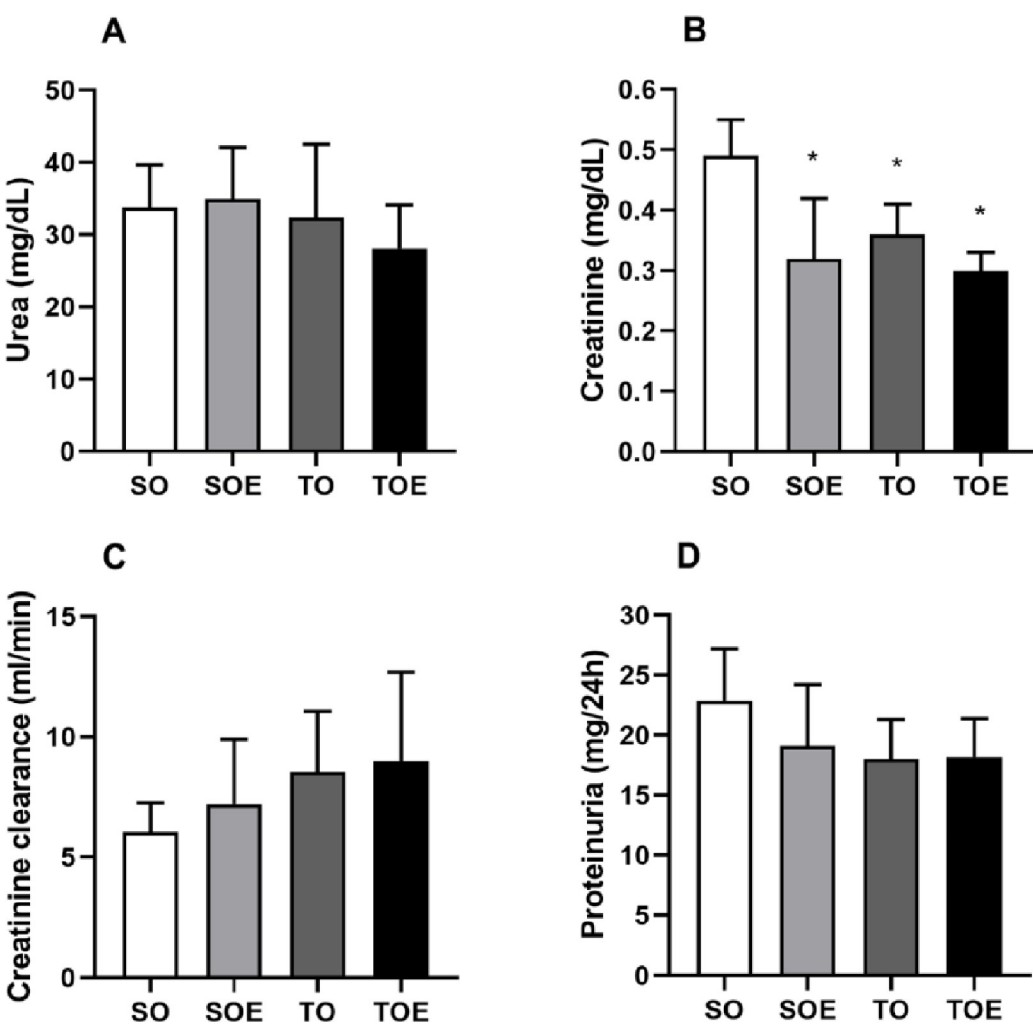

**Fig 2. Renal function parameters.** (A) Serum urea, (B) serum creatinine, (C) creatinine clearance and (D) protein levels in the urine in sedentary ovariectomized (SO), sedentary ovariectomized treated with enalapril (SOE), trained ovariectomized (TO) and trained ovariectomized treated with enalapril (TOE) groups. Data are presented as mean ± standard deviation. * p≤ 0.05 vs. SO.

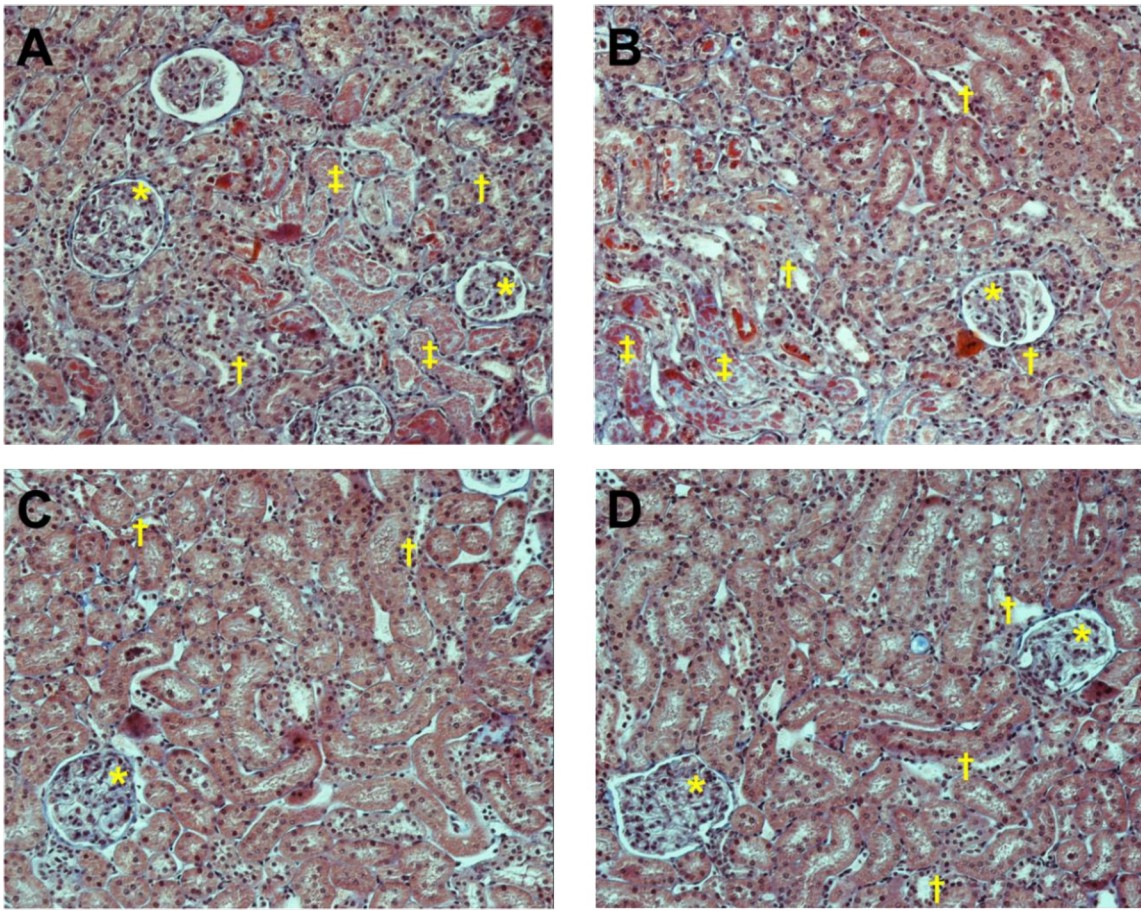

**Fig 3. Representative renal histological sections from the studied groups stained with Masson's trichrome at a 20x objective lens magnification.** Collagen deposition represented by blue color. (A) sedentary ovariectomized (SO), (B) sedentary ovariectomized treated with enalapril (SOE), (C) trained ovariectomized (TO) and (D) trained ovariectomized treated with enalapril (TOE) groups. * Glomeruli; † Tubular atrophy; ‡ Severe tubular injury.

In the tubulointerstitial injury analyses, the fields were classified into three ranges of morphological alterations: 0–25%, 26–50% and 51–100%. The Fig 3 displays fields of representative renal histological sections from the groups, highlighting areas of glomeruli, tubular atrophy, and severe tubular injury. After quantifying the morphological changes in renal tissue, it was observed a higher number of tubulointerstitial alterations in the SO group compared with treated groups. Specifically, the SOE, TO and TOE groups presented a higher number of fields with lower degree of morphological alteration (0–25%) compared with SO group. Additionally, the TO group had an additional increase in the 0–25% alterations when compared with SOE group. No differences were observed between the groups for morphological alteration between 26–50%. In addition, the SOE, TO and TOE groups presented lower number of fields with higher degree of morphological alteration (51–100%) in relation to the SO group. Trained group presented fewer fields with tubulointerstitial alterations between 51–100% compared with SOE and TOE groups. In turn, enalapril plus CET group presented fewer fields with higher degree of morphological alteration when compared with their respective sedentary group (SOE) (Fig 4).

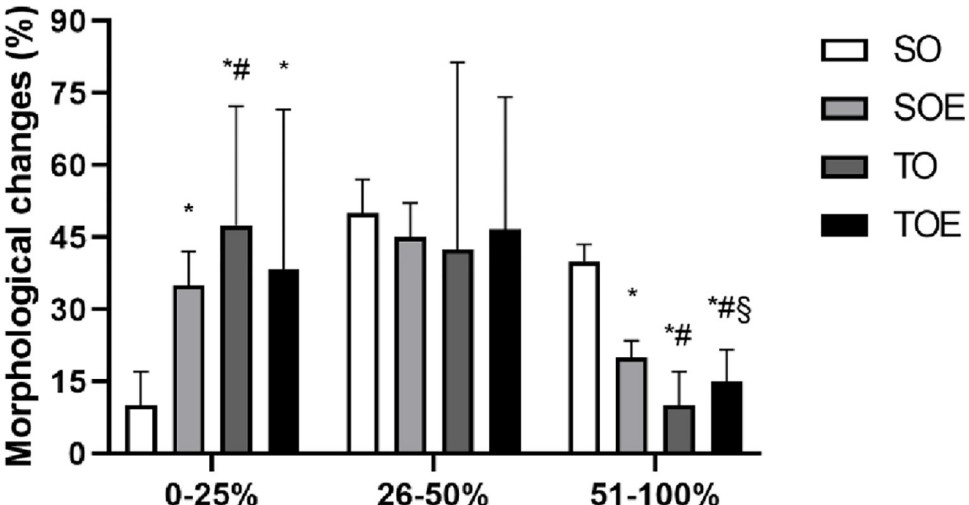

**Fig 4. Morphological tubulointerstitial alterations in renal tissue.** Analysis of 40 fields in the SO, SOE and TO groups and 60 fields in the TOE group. Total area of each field: 277.000 μm². Degree of morphological alteration: 0–25%; 26–50% and 51–100%. Data are presented as mean ± standard deviation. * p ≤ 0.05 vs. SO; # p ≤ 0.05 vs. SOE; § p ≤ 0.05 vs. TO. SO, sedentary ovariectomized, SOE, sedentary ovariectomized treated with enalapril, TO, trained ovariectomized; TOE, trained ovariectomized treated with enalapril.

## Renal inflammation and oxidative stress

No differences were found for TNF-α levels between groups. In turn, SOE, TO and TOE groups presented reduced levels of IL-6 compared with SO group. In addition, enalapril combined with CET promoted increase IL-10 levels compared with SOE and TO groups (Table 3).

Both trained groups showed reduced levels of lipoperoxidation compared with SO group; and the combination of enalapril plus CET promoted lower levels of TBARS in relation to their sedentary group (SOE). In addition, oxidized proteins (presented by carbonyls levels) were higher in SO compared with other groups. The $H_2O_2$ concentration, an important pro-oxidant, was reduced in the SOE group compared with SO group. Moreover, both trained

**Table 3. Inflammatory mediators and oxidative stress markers in renal tissue in the groups studied.**

|  | SO | SOE | TO | TOE |
|---|---|---|---|---|
| TNF-α (pg/mg protein) | 92.1 ± 13.9 | 81.9 ± 16.7 | 88.1 ± 13.0 | 97.5 ± 13.8 |
| IL-6 (pg/mg protein) | 311.1 ± 29.4 | 270.5 ± 26.0* | 216.4 ± 25.0*# | 238.3 ± 28.0*# |
| IL-10 (pg/mg protein) | 88.0 ± 5.4 | 80.6 ± 7.6 | 76.9 ± 8.6 | 97.2 ± 21.0*§ |
| TBARS (μmol/mg protein) | 2.08 ± 0.48 | 1.84 ± 0.35 | 1.54 ± 0.28* | 1.37 ± 0.26*# |
| Carbonyls (nmol/mg protein) | 5.35 ± 0.58 | 4.22 ± 0.36* | 4.32 ± 0.49* | 4.29 ± 0.35* |
| Hydrogen peroxide (μM) | 3.70 ± 0.19 | 2.73 ± 0.70* | 1.72 ± 0.53*# | 1.89 ± 0.40*# |
| SOD (USOD/mg protein) | 6.99 ± 0.25 | 8.68 ± 1.37* | 9.79 ± 0.90* | 9.24 ± 0.83* |
| CAT (nmol/mg protein) | 2.16 ± 0.30 | 2.4 ± 0.59 | 3.71 ± 0.49*# | 3.75 ± 0.44*# |

Data are presented as mean ± standard deviation.

SO, sedentary ovariectomized; SOE, sedentary ovariectomized treated with enalapril; TO, trained ovariectomized; TOE, trained ovariectomized treated with enalapril.

* p≤ 0.05 vs. SO

# p≤ 0.05 vs. SOE

§ p≤ 0.05 vs. TO. TNF-α, tumor necrosis factor alpha; IL-6, interleukin 6; IL-10, interleukin 10; TBARS, thiobarbituric acid reactive substances; SOD, superoxide dismutase; CAT, catalase.

groups showed reduced levels of $H_2O_2$ in relation to sedentary groups. Regarding antioxidants, SOE, TO and TOE groups showed increase in SOD activity (vs. SO); and only the trained groups presented higher CAT activity compared with sedentary groups (Table 3).

## Discussion

The beneficial effects of exercise training on AH have been widely demonstrated. The present study showed that the antihypertensive pharmacological treatment using enalapril plus CET was effective in reducing BP and attenuating vascular autonomic dysfunctions and baroreflex sensitivity in ovariectomized SHR. In addition, the combination of both therapies favourably modulated the inflammatory and oxidative stress renal profiles, reducing kidney injury and improving renal function.

Some factors such as the intensity of the exercise training protocol and the possible resistance to changing body weight could partially explain the absence of difference in body weight among the studied groups. Firstly, the exercise training performed at a higher relative intensity by the TO and TOE groups could lead to greater muscle mass gain. Although energy expenditure may have been higher in these groups, the potential increase in muscle mass induced by exercise training may not have resulted in differences in body weight among the studied groups, suggesting a possible overlapping effect. On the other hand, we hypothesized that our strain was impacted by the resistance in change body weight. SHR is a strain that exhibits greater resistance to changes in body weight compared to other strains (such as Wistar), even when food intake is similar [36].

Aerobic and resistance training performed on the same day is one of the main protocols of exercise training usually performed in clinical practice. In turn, we showed that 8 weeks of CET improved the performance of the trained groups (TO and TOE) compared to their respective sedentary groups (SO and SOE). Additionally, the group TOE demonstrated an additional improvement in performance in relation to the TO group in both final tests, suggesting that the treatment with enalapril could contribute for a higher enhancement of exercise capacity, further highlighting the importance of combining approaches in the treatment of AH.

Previous studies of our group have shown that ovariectomy induces additional BP elevation in SHR [15, 23], and that the exercise training is able to attenuate BP increase after ovariectomy in normotensive and hypertensive rats [37]. Shimojo et al. demonstrated that the combination of aerobic and resistance training performed on alternate days promoted a reduction in the mean BP of approximately 12% in SHR submitted to ovarian hormone deprivation compared with sedentary ovariectomized rats [15]. Recently, CET alone or combined with hydrochlorothiazide promoted similar hypotensive effect in female SHR [38]. In the present study, enalapril alone, CET alone or the combination of both antihypertensive therapies were efficient to promote reduction in the SBP ovariectomized SHR. Our findings highlight the equivalence of the proposed exercise training protocol in inducing chronic BP reduction similar to enalapril treatment.

Ovariectomy-induced high BP is accompanied by a large increase in vascular sympathetic modulation [39]. In our study, the absence of changes in VAR-SBP and LF-SBP in the SOE group suggests that the reduction in BP in SOE group is associated with other BP regulatory mechanisms. In turn, a reduced sympathetic influence has been observed using ACE inhibitors [40], being related to modulatory effect promoted by angiotensin-(1–7) on baroreflex sensitivity [41]. It well recognized that baroreflex sensitivity is an important predictor of mortality. Baroreflex dysfunction in fructose-induced AH tends to precede increased inflammation and oxidative stress damage in target organs, such as cardiac tissue [42]. In turn, both aerobic and

resistance exercises improve baroreflex sensitivity and cardiac sympathovagal balance [39], and the combination of both types of exercise was able to prevent alpha index dysfunction and hemodynamic, inflammatory and oxidative stress impairments [43]. In the present study, the CET associated with pharmacological treatment promoted a significant increase in spontaneous baroreflex sensitivity, expressed by the alpha index, compared with SO, SOE and TO groups (Fig 2D). It has been suggested that the reduction in BP *per se* after enalapril treatment is responsible to the normalization of the baroreceptor reflex [10]. Although enalapril therapy presented satisfactory BP reduction, only the CET was able to attenuate cardiovascular autonomic dysfunctions associated with AH and postmenopause, improving all autonomic parameters evaluated (VAR-SBP, SD-SBP, LF-SBP and alpha index). Moreover, the combination of enalapril with CET enhanced its protective effects, as observed in the VAR-SBP measurements and alpha index (Fig 1C and 1E). In accordance, similar results have been described in the literature. Maida et al. [44] identified increased baroreflex sensitivity and decreased LF-SBP in SHR when associated enalapril with 10-week swimming training. These authors hypothesized that exercise training could attenuate the activity of the angiotensin enzyme, thus decreasing angiotensin II and optimizing the action of enalapril.

Clinically, exercise has an important benefits in patients with chronic renal failure, such as decreasing BP, increasing tolerance to exercise and reducing pain, thus promoting better quality of life [45, 46]. However, the effects of exercise on serum creatinine and urea levels are still not well understood. There was evidence that exercise 5 days a week for 4 weeks promote reduction of creatinine in addition to a reduction in proteinuria in a model of acute renal failure [47]. In addition, a 12-week aerobic exercise program performed in moderate intensity was effective in decreasing BP, creatinine and urea levels in SHR, attenuating renal dysfunction [48]. In contrast, meta-analyses investigating the effects of exercise training in adults with CKD showed no changes in serum creatinine levels [45, 46]. In the present study, the groups that received intervention showed a trend of improvement in creatinine clearance and proteinuria compared to the SO group, although statistically not significant. We have considered three hypotheses that could explain our findings. Firstly, despite observing renal damage when evaluated histologically, we believe that this may not have been sufficient to induce changes in the assessed clinical markers. For example, proteinuria is a consequence of the progression of renal injury and occurs later in response to vascular and glomerular changes [49]. Additionally, we believe that the absence of changes in clinical markers may be related to the protective effect induced by the individual or combined approaches. Considering that AH is a relevant factor that triggers fibrosis, glomerulosclerosis, and proteinuria [50], we believe that chronic BP reduction in the SOE, TO, and TOE groups protected against the progression of these alterations at the end of the evaluation period. Finally, the female sex could be a protective factor to delayed the progression of the renal damage. Recently, Ansari et al. [51] showed a sex and age-related progression of renal fibrosis. Specifically, no differences in renal fibrosis indices were observed between female rats at 6.5 and 8 months of age. In contrast, female showed a protective effect from renal damage in relation to age-matched males [51].

Huang et al. has reported a greater area of renal fibrosis in SHR compared with normotensive Wistar-Kyoto rats [52]. Fibrosis in the renal tissue has been related with the increase of pro-inflammatory cytokines, especially IL-6 [52]. However, 12-week aerobic training attenuated morphological changes, but did not completely reverse them [52]. In our study, although we did not observe changes in the number of glomeruli, it was possible to identify tubulointerstitial morphological alterations in all studied groups by histological analyses (Fig 3). Tubulointerstitial alterations were more severe in the SO group, presenting 40% of their fields with 51–100% of morphological alteration, while SOE, TO and TOE groups presented 20, 10 and 15% of the fields, respectively (Fig 4). Tubular atrophy is evident around 30–60 weeks of age in

SHR [53, 54]. These results of renal histology corroborate with our findings of renal function since the same groups (treated with enalapril alone or combined with CET), which presented better renal function also presented lower renal tissue injury. Interesting, the group TO presented a better profile of tubulointerstitial alteration, regarding the quantity of fields with 0–25% and 51–100%, when compared with SOE and TOE.

Our group had verified that exercise training had potential to attenuate inflammation present in AH in ovariectomized rats, decreasing IL-6 levels in addition to TNF-α in the kidneys and heart [15], as well as reducing IL-10/TNF-α in heart when combined with antihypertensive pharmacological treatment [38]. Here, we did not observe differences in TNF-α levels in renal tissue. However, the SOE, TO and TOE groups showed lower renal IL-6 levels compared with SO group. Huang et al. demonstrated reduction of IL-6 in SHR submitted to exercise training, thus attenuating renal fibrosis [52]. These findings are extremely relevant since IL-6 plays an important role in CKD, with its production being induced by angiotensin II [55]. On the other hand, our protocol of CET combined with enalapril reduced renal oxidative stress damage, as demonstrated by reduced lipoperoxidation (TBARS) and protein oxidation (carbonyls). In addition, the pro-oxidant agent ($H_2O_2$) was reduced and the antioxidant enzymes were increased in renal tissue of the TOE group. We suggest that these results are in according to improvement in VAR-SBP due to recovery of vascular autonomic control, and in both pro and anti-inflammatory cytokines.

## Conclusions

Our findings showed that the combination enalapril plus CET enhances baroreflex sensitivity and sympathetic modulation, as well as reduces kidney inflammation and oxidative stress. Together, these improvements resulted in less renal tissue damage in hypertensive rats. These findings reinforce the preventive and therapeutic role played by exercise on renal tissue, impacting on improvement of the function of this important target organ.

## Author Contributions

**Conceptualization:** Michel Pablo dos Santos Ferreira Silva, Maycon Junior Ferreira, Kátia De Angelis.

**Data curation:** Michel Pablo dos Santos Ferreira Silva, Tânia Plens Shecaira, Danielle da Silva Dias, Débora Conte Kimura.

**Formal analysis:** Michel Pablo dos Santos Ferreira Silva, Maycon Junior Ferreira, Tânia Plens Shecaira, Débora Conte Kimura, Guiomar Nascimento Gomes.

**Investigation:** Michel Pablo dos Santos Ferreira Silva, Maycon Junior Ferreira.

**Methodology:** Michel Pablo dos Santos Ferreira Silva, Maycon Junior Ferreira, Danielle da Silva Dias.

**Project administration:** Kátia De Angelis.

**Supervision:** Kátia De Angelis.

**Validation:** Kátia De Angelis.

**Visualization:** Kátia De Angelis.

**Writing – original draft:** Michel Pablo dos Santos Ferreira Silva, Tânia Plens Shecaira, Maria Cláudia Irigoyen, Kátia De Angelis.

**Writing – review & editing:** Michel Pablo dos Santos Ferreira Silva, Maycon Junior Ferreira, Maria Cláudia Irigoyen, Guiomar Nascimento Gomes, Kátia De Angelis.

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
