## [Decision Letter · Decision Letter 0]

14 Sep 2023

PONE-D-23-12770Impact of exercise training associated with enalapril treatment on blood pressure variability and renal dysfunctions in an experimental model of arterial hypertension and menopausePLOS ONE

Dear Dr. De Angelis,

Thank you for submitting your manuscript to PLOS ONE. After careful consideration, we feel that it has merit but does not fully meet PLOS ONE’s publication criteria as it currently stands. Therefore, we invite you to submit a revised version of the manuscript that addresses the points raised during the review process.

We look forward to receiving your revised manuscript.

Kind regards,

Christopher Torrens

Academic Editor

PLOS ONE

Journal Requirements:

"KDA and MCI: This study was supported by National Council for Scientific and

Technological Development (CNPq) (407398/2021-0; 406792/2022-4)

(https://www.gov.br/cnpq/pt-br). Kátia De Angelis and Maria-Claudia Irigoyen are

recipients of CNPq Fellowship (CNPq-BPQ)."

Reviewers' comments:

Reviewer's Responses to Questions

**Comments to the Author**

1. Is the manuscript technically sound, and do the data support the conclusions?

Reviewer #1: Yes

Reviewer #2: Yes

2. Has the statistical analysis been performed appropriately and rigorously? 

Reviewer #1: Yes

Reviewer #2: Yes

3. Have the authors made all data underlying the findings in their manuscript fully available?

Reviewer #1: Yes

Reviewer #2: Yes

4. Is the manuscript presented in an intelligible fashion and written in standard English?

Reviewer #1: Yes

Reviewer #2: Yes

5. Review Comments to the Author

Reviewer #1: I do not find this paper so innovative in this field of research.

Many previous studies have shown the protective effects of both enalapril and physical activity on hypertension and menopause. It is absolutely reasonable thinking that these effects combined should be protective. However, I would not diminish the effort of the authors to conduct this study. The following points represent flaws, and need to be addressed:

1. In Materials and Methods section, please add how did You sacrificed the animals.

2. Histopathological figures are blurry, unclear, they need to be marked and described more clearly.

3. The discussion is written too extensively, it is necessary to shorten it, or even rewrite it. It is unnecessary to repeat all the obtained results again in the discussion. You have already described the results in the results section. You need to explain the obtained results. The literature used for the discussion is outdated. Please add more recent references. Also, the explanation of changes in serum creatinine values, in relation to normal creatinine clearances, is a bit speculative.

4. The paper has potentially inappropriate self-citations by authors.

Reviewer #2: Interesting study focused on the effect of exercise combined with antihypertensive treatment with ACEinhibitor (enalapril) on renal and vascular system in hypertensive female rats after ovariectomy. The conclusion was that combination of enalapril with exercise enhanced baroreflex sensitivity and sympathetic modulation and reduced inflammation and oxidative stress. These improvements resulted in less renal tissue damage.

Major comments:

1. Are there any data available about cardiac/vascular (aorta) tissue with regard to changes caused by hypertension and intervention under study after ovariectomy?

2. The age of ovariectomy is missing (quite important)

3. It should be discussed why have authors chosen for ovariectomy the 60th day.

Minor comments:

1. Enalapril is already not generally recommended drug (now perindopril ...) - page 5 - and the same sentence - typo - was chose x chosen

2. Did authors measured estradiol, testosterone to check real effect of ovariectomy and sex-hormone changes?

3. The weight was not different between groups including "exercise" groups - and according to authors there were no differences in water, food intake - but suppose in exercising rats there was increased energy expedinture - is there another explanation?

4. Some graphs/pictures could make the paper more attractive

In conclusion interesting experimental study focused on the effect of intervention composed of exercise and antihypertensive pharmacotherapy after ovariectomy in hypertensive strain of rats. Some points are to be addressed in more detail/added/discussed.

6. PLOS authors have the option to publish the peer review history of their article (what does this mean?). If published, this will include your full peer review and any attached files.

Reviewer #1: No

Reviewer #2: No

---

## [Author Response · Author response to Decision Letter 0]

8 Nov 2023

The authors would like to thank the reviewers for their valuable comments. We carefully considered all of the suggestions and accomplished with all of the requests within the aim and possibilities of the present study. Responses to each question are provided below, and changes made in the manuscript are highlighted.

Reviewer #1: 

I do not find this paper so innovative in this field of research.

Many previous studies have shown the protective effects of both enalapril and physical activity on hypertension and menopause. It is absolutely reasonable thinking that these effects combined should be protective. However, I would not diminish the effort of the authors to conduct this study. The following points represent flaws, and need to be addressed:

Thank you for initial considerations and for opportunity to discuss this manuscript. 

1. In Materials and Methods section, please add how did you sacrificed the animals.

We thank the reviewer for the observation. We have added a detailed description of this procedure in the 'Tissue collection' section, as requested. This change is highlighted in the manuscript. 

2. Histopathological figures are blurry, unclear, they need to be marked and described more clearly.

We have readjusted the histological figures and described them in a more appropriate manner, as requested. Specifically, the image has been reinserted in the best available quality. Furthermore, we have increased the size of the labels in each image (A - D), appropriately described their meanings in the Figure 3 caption, and reformulated the discussion of the observed results. The changes are highlighted in the manuscript.

3. The discussion is written too extensively, it is necessary to shorten it, or even rewrite it. It is unnecessary to repeat all the obtained results again in the discussion. You have already described the results in the results section. You need to explain the obtained results. The literature used for the discussion is outdated. Please add more recent references. Also, the explanation of changes in serum creatinine values, in relation to normal creatinine clearances, is a bit speculative.

We thank the reviewer for the valuable suggestion. As suggested, we have rewritten the discussion and included new references that reinforce our findings and hypotheses raised in the study. The changes are highlighted in the manuscript.

4. The paper has potentially inappropriate self-citations by authors.

We thank the reviewer for this important observation. We have reviewed the manuscript and adjusted the references as requested.

Reviewer #2:

Interesting study focused on the effect of exercise combined with antihypertensive treatment with ACE inhibitor (enalapril) on renal and vascular system in hypertensive female rats after ovariectomy. The conclusion was that combination of enalapril with exercise enhanced baroreflex sensitivity and sympathetic modulation and reduced inflammation and oxidative stress. These improvements resulted in less renal tissue damage.

Major comments:

1. Are there any data available about cardiac/vascular (aorta) tissue with regard to changes caused by hypertension and intervention under study after ovariectomy?

We thank you for initial considerations and for opportunity to discuss this manuscript.

Recently, our group demonstrated beneficial results on cardiac tissue by combining pharmacological and non-pharmacological approaches. Specifically, we observed an increase in IL-10 and the IL-10/TNF ratio in rats subjected to hydrochlorothiazide and concurrent exercise training compared to ovariectomized control rats and ovariectomized rats treated with the drug alone. Additionally, the combination of these approaches resulted in a greater magnitude of beneficial adaptation on oxidative stress markers than pharmacological treatment alone (DOI: 10.1371/journal.pone.0289715). Considering our comment, we added this reference in the discussion. This change is highlighted in the manuscript.

Currently, we have results in the publication phase demonstrating similar benefits for trained rats treated with enalapril in cardiac tissue, such as a reduction in TNF-alfa and oxidative profile (hydrogen peroxide), as well as an improvement in antioxidant defense (catalase and glutathione peroxidase). Therefore, the findings in renal tissue observed in the present study have also been confirmed in cardiac tissue, which reinforces our hypothesis of inducing additional benefits when combining these approaches. 

2. The age of ovariectomy is missing (quite important).

Thank you for important observation. We have added the age of ovariectomy in the 'Ovariectomy' section. This change is highlighted in the manuscript.

3. It should be discussed why have authors chosen for ovariectomy the 60th day.

The surgical removal of both ovaries from each female rat was performed at three months of age (90 days) (as insert in the ‘Ovariectomy section’). At this age, the factor of age/aging is eliminated, allowing us to specifically identify the impacts caused by ovarian hormonal deprivation. Cessation of ovarian hormones remained until the end of the protocol (approximately 60 days post-ovariectomy, involving an 8-week period with either a placebo or interventions, followed by subsequent hemodynamic assessments). Taking this into consideration, we have revised the sentence to clarify this matter. The revised sentence has been added to the ‘Hemodynamic and autonomic assessments’ section. The changes are highlighted in the manuscript.

Minor comments:

1. Enalapril is already not generally recommended drug (now perindopril ...) - page 5 - and the same sentence - typo - was chose x chosen.

We thank the reviewer for these comments. In fact, recent advancements in drug development have yielded increasingly favorable outcomes, enabling more targeted pharmacological prescriptions. Significantly, the prescription of enalapril as a common antihypertensive treatment in developing countries, such as in Brazil (DOI: 10.1590/S1518-8787.2016050006154). Brazil's policies promoting free access to medications have expanded and serve as exemplars of public health initiatives. In this regard, the current healthcare policy governing the Unified Health System (Sistema Único de Saúde [SUS]) in Brazil facilitates cost-free access, upon a medical prescription, to affordable antihypertensive drugs like enalapril for diagnosed hypertensive patients, thereby enhancing societal accessibility to prescribed pharmacological treatments.

Considering the reviewer comments, we have appropriately corrected the typing error in the sentence.

2. Did authors measured estradiol, testosterone to check real effect of ovariectomy and sex-hormone changes?

The effect of ovarian hormonal deprivation has been extensively demonstrated, including for our research group. Specifically, ovariectomy is a procedure that has been systematically performed by our research group. We have previously presented information on the ovarian hormonal profile of female rats at different stages of the estrous cycle, as well as following ovariectomy. Using immunoassay methods, we have observed higher estrogen concentrations during the diestrus phase compared to the estrus phase, with estrogen levels below detectable limits in males and ovariectomized females (DOI: 10.1186/s13293-020-00290-y). These findings are consistent with prior data published (10.1016/j.bbr.2011.10.047; DOI: 10.1002/ar.21247; 10.1590/S1807-59322010000700009; DOI: 10.1016/j.maturitas.2009.11.007; DOI: 10.1097/gme.0b013e3182358c9c). Considering our observation, we added information regarding estradiol levels after ovariectomy in the “Ovariectomy” item of the methods section. The ovariectomy was conducted by a professional experienced in this procedure in the present study.

Unfortunately, we did not have additional plasma samples to analyze androgen levels in the present study. 

3. The weight was not different between groups including "exercise" groups - and according to authors there were no differences in water, food intake - but suppose in exercising rats there was increased energy expenditure - is there another explanation?

We have demonstrated that ovariectomy results in a greater weight gain after 8 weeks when compared to non-ovariectomized rats, thus indicating a possible effect of ovarian hormone deprivation on anthropometric profiles. Interestingly, this difference also persists after treatment with a diuretic or exercise training (DOI: 10.1371/journal.pone.0289715). We believe that the involvement of the Renin-Angiotensin-Aldosterone System could be one of the possible mechanisms related to the absence of anthropometric changes in hypertensive rats. It has been suggested that the production of angiotensinogen by adipose tissue promotes adipogenesis, and angiotensin II can induce the process of adipose tissue accumulation by increasing the production of key lipogenic enzymes and inducing lipogenesis through the AT2 receptor pathway (DOI: 10.1002/cphy.c160031). 

Regarding the present study, some factors such as the intensity of the exercise training protocol and the possible resistance to changing body weight could partially explain the absence of difference in body weight among the studied groups. Firstly, the exercise training performed at a higher relative intensity by the TO and TOE groups could lead to greater muscle mass gain. Although energy expenditure may have been higher in these groups, the potential increase in muscle mass induced by exercise training may not have resulted in differences in body weight among the studied groups, suggesting a possible overlapping effect. On the other hand, we hypothesized that our experimental model (SHR) was impacted by the resistance in change body weight. SHR is a strain that exhibits greater resistance to changes in body weight compared to other strains (such as Wistar), even when food intake is similar (DOI: 10.1016/0091-3057(94)90498-7). Furthermore, although weighing the food is a widely adopted measure to establish daily consumption, it is a method that has limitations and may not accurately reflect the actual amount of food ingested. 

4. Some graphs/pictures could make the paper more attractive.

We thank the reviewer for this valuable suggestion. We have reworked the graphs and figures and included them in this revised version of the manuscript.

In conclusion interesting experimental study focused on the effect of intervention composed of exercise and antihypertensive pharmacotherapy after ovariectomy in hypertensive strain of rats. Some points are to be addressed in more detail/added/discussed.

We thank the reviewer for the valuable insights into our study. As requested, we have adjusted and addressed our findings in a more appropriate manner.

---

## [Decision Letter · Decision Letter 1]

18 Dec 2023

Impact of exercise training associated with enalapril treatment on blood pressure variability and renal dysfunctions in an experimental model of arterial hypertension and postmenopause

PONE-D-23-12770R1

Dear Dr. De Angelis,

We’re pleased to inform you that your manuscript has been judged scientifically suitable for publication and will be formally accepted for publication once it meets all outstanding technical requirements.

Kind regards,

Christopher Torrens

Academic Editor

PLOS ONE

Additional Editor Comments (optional):

Reviewers' comments:

Reviewer's Responses to Questions

**Comments to the Author**

1. If the authors have adequately addressed your comments raised in a previous round of review and you feel that this manuscript is now acceptable for publication, you may indicate that here to bypass the “Comments to the Author” section, enter your conflict of interest statement in the “Confidential to Editor” section, and submit your "Accept" recommendation.

Reviewer #1: All comments have been addressed

Reviewer #2: All comments have been addressed

2. Is the manuscript technically sound, and do the data support the conclusions?

Reviewer #1: Yes

Reviewer #2: Yes

3. Has the statistical analysis been performed appropriately and rigorously? 

Reviewer #1: Yes

Reviewer #2: Yes

4. Have the authors made all data underlying the findings in their manuscript fully available?

Reviewer #1: Yes

Reviewer #2: Yes

5. Is the manuscript presented in an intelligible fashion and written in standard English?

Reviewer #1: Yes

Reviewer #2: Yes

6. Review Comments to the Author

Reviewer #1: The authors successfully responded to all objections. The authors significantly improved their nanuscript .In this form, I think that this manuscript should be accepted.

Reviewer #2: I am satisfied with the answers and improvements made by authors, including substantial changes in Discussion.

7. PLOS authors have the option to publish the peer review history of their article (what does this mean?). If published, this will include your full peer review and any attached files.

Reviewer #1: No

Reviewer #2: No

---

## [Editor Report · Acceptance letter]

2 Jan 2024

PONE-D-23-12770R1 

PLOS ONE

Dear Dr. De Angelis, 

I'm pleased to inform you that your manuscript has been deemed suitable for publication in PLOS ONE. Congratulations! Your manuscript is now being handed over to our production team.

Kind regards, 

on behalf of

Dr. Christopher Torrens 

Academic Editor

PLOS ONE